# Influence of Ingestion of Lactulose on γ-Lactones Emanating from Human Skin Surface

Yoshika Sekine [1,*], Shiori Uchiyama [1], Michihito Todaka [2], Yohei Sakai [3], Ryo Sakiyama [3], Hiroshi Ochi [3], Maho Muramatsu [1], Satomi Asai [4] and Kazuo Umezawa [5]

[1] Graduate School of Science, Tokai University, 4-1-1 Kitakaname, Hiratsuka 259-1292, Japan
[2] AIREX Inc., 4-1-1 Kitakaname, Hiratsuka 259-1292, Japan
[3] Food Ingredients & Technology Institute, R&D Division, Morinaga Milk Industry Co., Ltd., 5-1-83, Higashihara, Zama 252-8583, Japan
[4] Department of Laboratory Medicine, School of Medicine, Tokai University, 143 Shimokasuya, Isehara 259-1193, Japan
[5] Department of Emergency and Critical Care Medicine, School of Medicine, Tokai University, 143 Shimokasuya, Isehara 259-1193, Japan
* Correspondence: sekine@keyaki.cc.u-tokai.ac.jp; Tel.: +81-463-58-1211 (ext. 3156)

**Abstract:** Lactulose is known to grow health-promoting bacteria, with an increase in the production of beneficial metabolites, such as lactic acid and short-chain fatty acids (SCFAs) in the colon. Ingestion of lactulose powder at a food dosage level is known to reduce the dermal emanation of ammonia, a typical human skin gas which potentially affects body odour. However, no study has reported the effect of lactulose on human skin gases other than ammonia. In this study, the influence of lactulose ingestion on the dermal emissions of γ-lactones, volatile cyclic esters with sweet smells, was investigated in healthy subjects. Healthy participants ingested the lactulose powder with a food dosage of 4 g d$^{-1}$ once a day for 2 weeks. γ-lactones emanating from the skin surface were collected from each participant's forearm by using a passive flux sampler, and six kinds of γ-lactones, namely, γ-hexalactone (C6), γ-heptalactone (C7), γ-octalactone (C8), γ-nonalactone (C9), γ-decalactone (C10), and γ-undecalactone (C11), were determined by gas chromatography–mass spectrometry. Quantification of bifidobacteria in faeces collected before and after ingestion of lactulose for 2 weeks was carried out by using real-time PCR. The results showed a significant increase in the dermal emission fluxes of sweet-smelling C10 and C11 lactones as the number of bifidobacteria increased in the faeces, presumably mediated by SCFAs produced in the colon.

**Keywords:** human skin gas; γ-lactones; prebiotic; lactulose; SCFAs

## 1. Introduction

Lactulose (4-O-β-D-galactopyranosyl-β-D-fructose) is a simple disaccharide made up of galactose and fructose produced by isomerisation of the milk sugar lactose [1,2]. Clinical doses of lactulose are used for the treatment of high ammonia levels in blood, which leads to hepatic encephalopathy and constipation [3]. Meanwhile, lactulose at low doses (<10 g d$^{-1}$) is used as a prebiotic which promotes an increase in the abundance of *Bifidobacterium* in the human gut and is thought to promote human health by increasing the production of beneficial metabolites such as lactic acid and short-chain fatty acids (SCFAs), including acetate, propionate, and butyrate in the colon [4,5]. Our previous study [6] demonstrated the influence of lactulose at a food dosage of 4 g d$^{-1}$ on ammonia emanating from the skin surface of healthy subjects. When ammonia is formed in the human body by internal metabolism, it rises to the skin surface with perspiration (sweat gland route) and/or directly in volatiles from the blood through cutaneous layers (blood route) [7,8]. This emanation can cause a characteristic pungent body odour [7,8]. A significant decrease in the emission of skin ammonia was observed eight days after lactulose ingestion, and



the flux of dermal emission of ammonia decreased as the number of faecal bifidobacteria grew [6]. However, no study has reported the influence of lactulose on human skin gases other than ammonia.

γ-lactones are volatile cyclic esters derived from fatty acid metabolism [9,10]. These compounds are naturally present in fruits, including peaches, plums, pineapples, and strawberries, and contribute to their complex aroma and flavour [10,11]. For example, γ-octalactone (C8), γ-nonalactone (C9), γ-decalactone (C10), and γ-undecalactone (C11) are key odorants in peach and apricot [12,13]. Biosynthetic studies indicate that several pathways originating from the β-oxidation of middle- and/or long-chain fatty acids, such as oleic acid, linoleic acid, and palmitoleic acid, are responsible for the structural diversity of lactones [14]. Recently, these γ-lactones have also been found in a trace gas mixture emanating from the human skin surface [15,16]. They contribute to sweet body scents, particularly in young females, because a greater dermal emission was found in teenage girls [15]. Meanwhile, SCFAs, metabolites of gut microbiota, have attracted increasing interest due to their effect on host adiposity and contribution to anti-obesity by increasing fatty acid oxidation [17,18]. Therefore, when fatty acid oxidation is enhanced by changes in the gut microbiota, dermal emission of γ-lactones can be affected.

In this study, we aimed to investigate the influence of oral ingestion of lactulose with a food dosage of 4 g d$^{-1}$ on the amount of sweet-smelling γ-lactones released from the skin surface of healthy subjects. The emission mechanism of γ-lactones was also discussed by measuring dermal emissions of SCFAs as their free acid forms.

## 2. Methods

### 2.1. Participant Tests

Open-label and before–after trials of lactulose on the dermal emission of γ-lactones and SCFAs were conducted by recruiting 6 healthy participants (5 males and 1 female; age: 22–23) with 169 ± 7.2 cm height, 59 ± 7.2 kg weight, and 21 ± 1.3 body mass index. A supplemental test was carried out to establish the relationship between cutaneous emissions of γ-lactones and the number of bifidobacteria to additionally recruit four healthy participants. The relationship was then investigated for a total of 10 participants (7 males and 3 females; age: 21–24) with 171 ± 7.7 cm height, 60 ± 7.2 kg weight, and 20 ± 2.4 body mass index. For the selection of the test participants, see the exclusion criteria described in our previous study [6].

The test schedule is shown in Figure 1. Four grams of lactulose powder MLC-97 (≥97%, Morinaga Milk Industry Co., Ltd., Tokyo, Japan) was provided to the healthy participants daily for 2 weeks. The intake time and methods were based on each participant's choice. Since the lactulose is soluble in water, it can be easily taken with water or other beverages. It can be also eaten with meals as a sweetener. During the test period, the participants were allowed to have their usual meals without any restrictions. Meanwhile, they were instructed to refrain from the use of supplements and pharmaceuticals that might affect intestinal microorganisms and defaecation in advance.

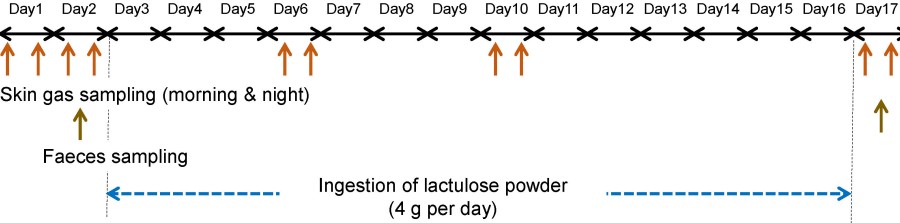

**Figure 1.** Schedule of skin gas and faeces samplings and ingestion of the tested disccharide powder.

Faecal samples were collected before and after the ingestion period (Day 2 and 17) to determine the number of bifidobacteria in faeces. The collected samples were immediately sent in frozen conditions (−18 °C) to the laboratory of Morinaga Milk Industry and stored

at −80 °C until use for real-time polymerase chain reaction (PCR). Sampling of the human skin gas was carried out in the morning (before breakfast, bathing, or taking a shower) and night (1 h before bed) on 5 specific days before, during, and after the ingestion period (Day 1, 2, 6, 10, and 17). No special surface treatment was conducted on the forearm prior to sampling. As for the additional 4 participants, skin gas sampling was conducted on Day 2 and 17 according to the date of faecal sampling. The human skin gas samples were stored in a household refrigerator, sent to the laboratory of Tokai University soon after Day 17, and analysed by gas chromatography–mass spectrometry (GC-MS).

### 2.2. Skin Gas Sampling and Analysis

The amount of dermal emission of human skin gases was determined in healthy participants using a passive flux sampler (PFS) in conjunction with GC-MS [15,16,19]. The target analytes were six kinds of γ-lactones, namely, γ-hexalactone (C6), γ-heptalactone (C7), γ-octalactone (C8), γ-nonalactone (C9), γ-decalactone (C10), and γ-undecalactone (C11), and SCFAs, such as acetic acid, propionic acid, and butyric acid.

Figure 2 shows a schematic drawing of the PFS (MonoTrap® SG DCC18) commercially available from GL Sciences, Tokyo, Japan. The tiny sampling device was composed of a glass vial, a polypropylene (PP) screw cap, a disk-type adsorbent material as trapping media, and a PTFE O-ring as a stopper. The PFS was gently fixed on the surface of the participant's forearm using a piece of medical tape for one hour.

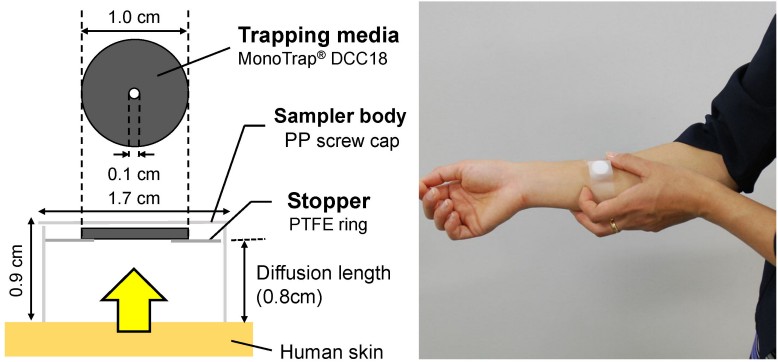

**Figure 2.** Schematic drawing of the PFS for the collection of γ-lactones and SCFAs. PFS was put on the surface of the non-dominant forearm of a healthy participant.

The trapped human skin gases were eluted into 500 μL dichloromethane with 15 min of ultrasonic extraction. The chromatographic analysis was performed for the extracts using a gas chromatograph model 7890B (Agilent Technologies, Santa Clara, CA, USA) and a mass selective detector, JMS-Q1050GC MkII (JEOL, Tokyo, Japan). One microlitre of sample extracts, blank extracts, and quantitation standards was injected with a split ratio of 35:1 into a DB-IMS capillary column (Agilent Technologies, Santa Clara, CA, USA), 30 m × 0.25 mm I.D. × 0.25 μm film thickness. The carrier gas was G1 grade helium (Taiyo Nippon Sanso, Tokyo, Japan) at a flow rate of 1.0 mL min⁻¹. The injector port was maintained at 280 °C. The oven temperature was programmed as follows: held at 50 °C for 8 min, ramped at 6 °C·min⁻¹ to 120 °C, and ramped at 20 °C·min⁻¹ to 280 °C and held for 2 min. Signals for sample extracts were acquired using a real-time Selected Ion Monitoring mode. The emission flux of the target human skin gas, $E$ (ng cm⁻² h⁻¹) was calculated using the following formula:

$$E = \frac{W}{S \cdot t} \tag{1}$$

where $W$ is the amount (ng) of collected human skin gases, $S$ is the effective cross-section of the adsorbent (0.594 cm²), and $t$ is the sampling duration (1.0 h). The emission fluxes of SCFAs were calculated as their free acid forms, which are the typical volatile species of SCFAs.

### 2.3. Determination of Faecal Bifidobacterial Numbers

Faecal bifidobacterial numbers were determined by real-time PCR. Bacterial DNA was extracted from faecal samples and amplified by quantitative PCR, as described in ref. [6]. The number of bifidobacteria per gram of faeces was quantified using *Bifidobacterium* genus-specific forward (5′ CTCCTGGAAACGGGTGG 3′) and reverse (5′ GGTGTTCTTCC-CGATATCTACA 3′) primers [20]. A dilution series of *Bifidobacterium longum* ATCC 15707T cells [21] was used for calibration.

### 2.4. Statistical Analyses

IBM SPSS Statistics 25 was used to perform the statistical analyses. Differences in the number of bifidobacteria before and after lactulose ingestion were analysed using the Wilcoxon signed-rank test. The increasing or decreasing trends in the dermal emission fluxes of γ-lactones and SCFAs were analysed using the Jonckheere trend test, also known as the Jonckheere–Terpstra test. Pearson's correlation coefficients were obtained between changes in the dermal emission flux of γ-lactones and the number of bifidobacteria. Unless indicated otherwise, statistical significance was set at $p < 0.05$.

## 3. Results

Figure 3 shows a comparison between the 16S rRNA gene copy numbers (GCNs) of bifidobacteria per gram of faeces before and after 2-week ingestion of lactulose powder at a dose of 4 g d$^{-1}$. The arithmetic mean and standard deviations are shown for six healthy participants recruited first. As can be predicted based on previous works, the common logarithm of GCNs g$^{-1}$ faeces significantly increased from $8.91 \pm 0.20$ to $9.27 \pm 0.37$ after the 2-week ingestion (Wilcoxon signed-rank test, $p = 0.046$).

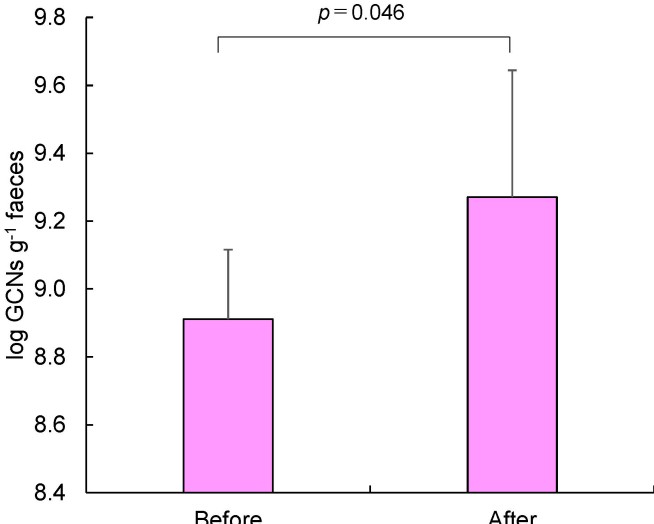

**Figure 3.** Comparison of the 16S rRNA gene copy numbers of bifidobacteria in faeces of 6 healthy participants before (Day 2) and after (Day 17) daily ingestion of lactulose for 2 weeks ($n = 6$).

Figure 4 shows the variations in the dermal emission fluxes of C6-C11 lactones and SCFAs measured for six healthy participants from their forearms. The average emission fluxes in the morning and at night on each sampling day were used as daily values, and the arithmetic mean and standard deviation of the daily values of the six participants are shown in this figure.

The dermal emission fluxes of γ-lactones before lactulose intake (Day 1 and 2) were in the order of C11 = C9 > C8 > C10 > C7 = C6. However, the order changed after the ingestion of lactulose powder (Day 6, 10, and 17). A remarkable increase in the cutaneous emission was observed for the C7, C10, and C11 lactones, resulting in the following order: C11 > C9 = C7 > C10 > C8 > C6. For example, the dermal emission fluxes of C11 lactone

at Day 1 was $1.6 \pm 0.77$ ng cm$^{-2}$ h$^{-1}$, and equivalent to $1.6 \pm 0.89$ ng cm$^{-2}$ h$^{-1}$ at Day 2. After the ingestion of the disaccharide powder, the flux increased to $3.0 \pm 1.3$ ng cm$^{-2}$ h$^{-1}$ at Day 6 (four days after the ingestion), $3.2 \pm 0.98$ ng cm$^{-2}$ h$^{-1}$ at Day 10 (eight days after the ingestion), and $3.2 \pm 0.91$ ng cm$^{-2}$ h$^{-1}$ at Day 17 (one day after the end of ingestion). A Jonckheere trend test was applied to the variations from Day 2 to Day 17. The results indicated significant increasing trends in the dermal emission fluxes of C7 lactone ($p = 0.022$) with approx. 2.8 times increase from Day 2 to Day 17, C10 lactone ($p = 0.025$) with approx. 1.8 times increase, and C11 lactone ($p = 0.002$) with approx. 2.0 times increase after oral intake of lactulose powder. No apparent trend was found in the cutaneous emissions of C6, C8, and C9 lactones. The reason is not clear at present for the differences in the trends of $\gamma$-lactones.

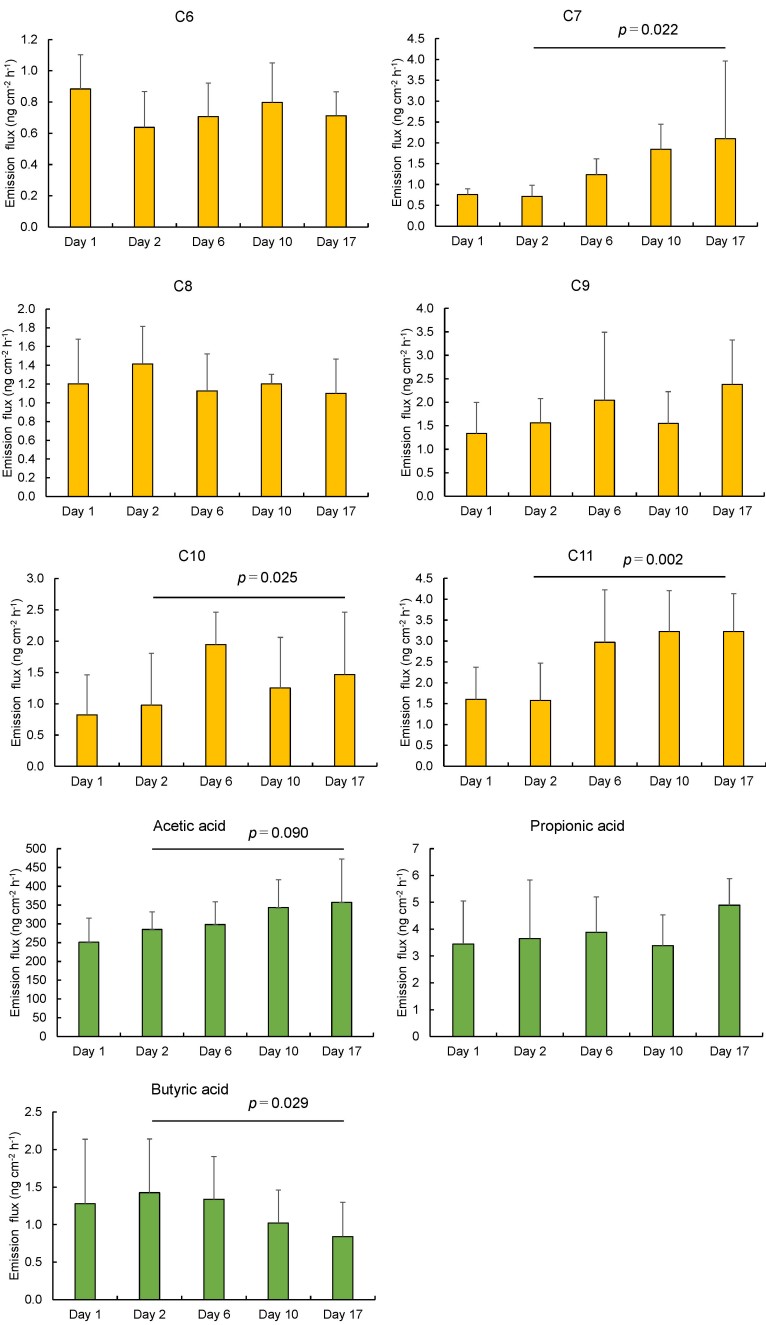

**Figure 4.** Variation in the fluxes of dermal emission of $\gamma$-lactones and SCFAs collected from the forearm of 6 healthy participants before (Day 1 and 2) and after (Day 6, 10, and 17) daily ingestion of lactulose. Jonckheere trend test was conducted for the values from Day 2 to Day 17.

On the other hand, the variation of dermal emission was different for SCFAs. The emission flux of acetic acid gradually increased from $2.8 \times 10^2 \pm 52$ ng cm$^{-2}$ h$^{-1}$ at Day 2 to $3.5 \times 10^2 \pm 1.8 \times 10^2$ ng cm$^{-2}$ h$^{-1}$ at Day 17 (Jonckheere trend test, $p = 0.09$). No apparent change was observed for propionic acid, whereas a significant decrease was found in the dermal emission flux of butyric acid from Day 2 to Day 17 ($p = 0.029$). However, the simple sum of cutaneous emissions of these three SCFAs showed an increasing tendency after ingestion of lactulose because the acetic acid emission was much greater than other two acids.

Figure 5 presents scatter diagrams of the 16S rRNA gene copy numbers of bifidobacteria in the faeces and the dermal emission fluxes of γ-lactones in all 10 healthy volunteers. The X-axis shows the changes in log GCNs g$^{-1}$ from Day 2 to Day 17, $\varDelta$log GCNs g$^{-1}$; the Y-axis represents the changes in dermal emissions from Day 2 to Day 17, $\varDelta E$ (ng cm$^{-2}$ h$^{-1}$). Significant correlations were observed for C10 lactone ($r = 0.73$, $p = 0.018$) and C11 lactone ($r = 0.74$, $p = 0.014$). A similar tendency was also observed for the C7 lactone.

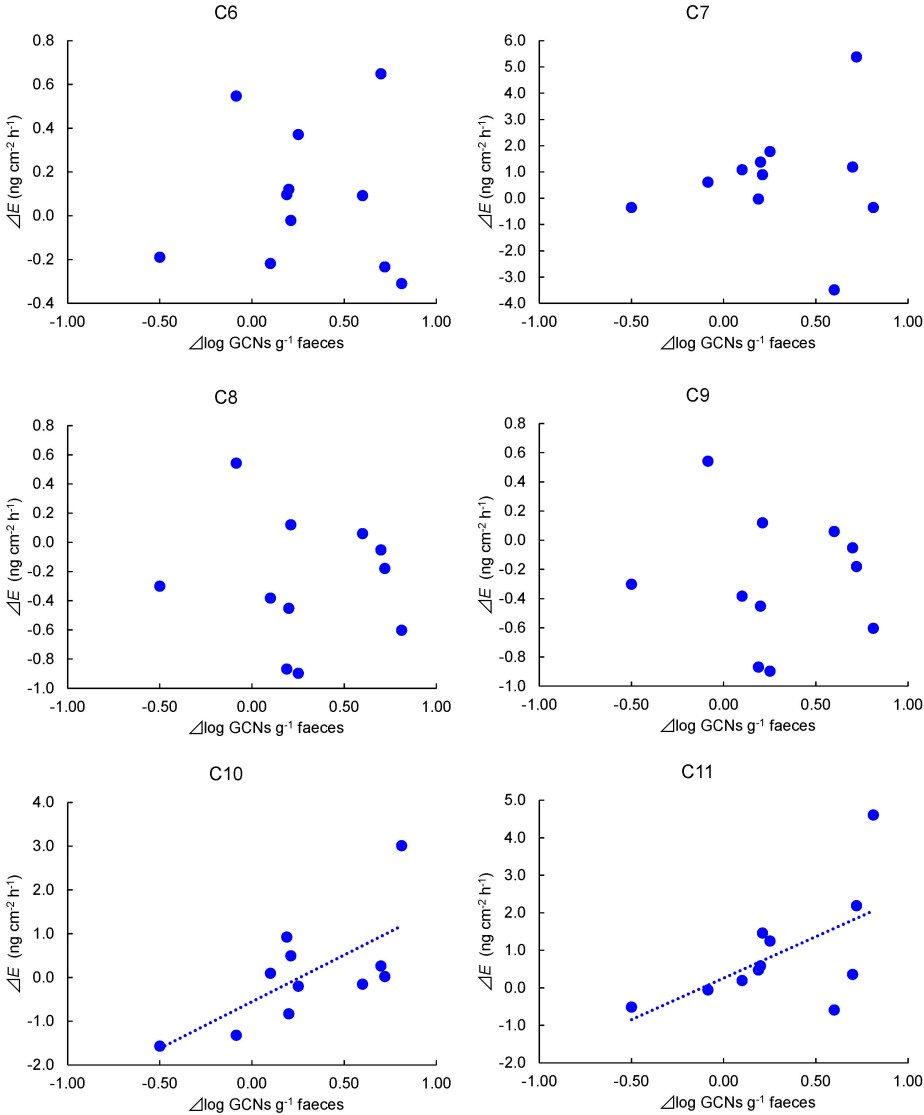

**Figure 5.** Scatter plots of the 16S rRNA gene copy numbers of bifidobacteria in faeces and dermal emission fluxes of γ-lactones in all 10 healthy participants (original 6 plus additional 4 participants). X-axis: changes in log GCNs g$^{-1}$ from Day 2 to Day 17; Y-axis: changes in the dermal emissions from Day 2 to Day 17.

## 4. Discussion

This study provides novel evidence for the influence of lactulose ingestion on sweet-smelling gases released from the human skin surface. The fluxes of dermal emission of C10 and C11 lactones increased as the number of bifidobacteria increased. C10 lactone has a strong peach-like odour, and C11 lactone has a coconut and/or fruity smell [22]. Lactulose products are commercially available in the form of syrup, beverages, and yoghurt. Thus, by integrating our previous knowledge of ammonia [6], the daily ingestion of lactulose probably contributes to improving body scent by increasing the dermal emission of sweet-smelling lactones and decreasing pungent odorous ammonia.

Willems et al. [16] reported an increase in the dermal emission of C11 lactone in aged British people after oral intake of New Zealand blackcurrant powder containing anthocyanin. However, the mechanism underlying dermal γ-lactone formation is not fully understood. One possible mechanism is that the microbiota form lactones on the human skin surface. *Malassezia furfur* (formerly known as *Pityrosporum*, a lipophilic yeast) commonly colonises the human skin surface, especially the scalp, and can cause mycosis and other adverse health disorders. Labows et al. [23] found that the genus *Malassezia furfur* produced volatile C8–C12 lactones during growth on culture media supplemented with human sebum. They also suggested that the production of γ-lactones may serve as a biomarker in the identification of the genus *Malassezia furfur* on the human scalp, where abundant serum and other nutrients [23] are present. However, it is difficult to link the fungal activity on the human skin surface and ingestion of lactulose, as well as New Zealand blackcurrant, due to a lack of sufficient evidence. Another possible mechanism is the promotion of fatty acid oxidation mediated by the SCFAs produced in the colon.

Acetate, propionate, and butyrate constitute 95% of the microbial-derived SCFAs [17]. After being produced by gut microbiota, SCFAs are initially used by intestinal epithelial cells as an energy source. Residues are subsequently transported to blood vessels in a concentration-dependent manner and then migrate to the liver and muscles through blood circulation [17]. SCFAs do not directly travel from the blood to the skin surface because they exist in non-volatile chemical forms such as acetate, propionate, and butyrate salts in weakly basic blood plasma. However, when they emerge with perspiration, an acidic environment on the skin surface (pH = 5–6) can release these SCFAs as free acids into the air, because their p$K$a is approximately 4.8. Therefore, the dermal emission of acetic acid appears to be a good biomarker for sweating [24]. The composition of SCFAs in the intestine is not necessarily reflected in the composition of dermal emission fluxes because there may be other potential emission routes of dermal SCFAs, such as bacterial activity on the skin and food metabolism. However, it should be noted that the composition of the three dermal SCFAs apparently changed after ingestion of lactulose, as shown in Figure 4, and the fluxes of cutaneous emission of acetic acid and/or the sum of the three SCFAs showed an increasing trend with elapsed days.

Recently, SCFAs have attracted increasing interest due to their effect on host adiposity and contribution to obesity by increasing fatty acid oxidation [17,18]. den Besten et al. [25] reported that dietary SCFAs can act as signalling molecules in the body, inducing a peroxisome proliferator-activated receptor-gamma (PPARγ)-dependent switch from lipid synthesis to lipid utilisation by enhancing oxidative metabolism; hepatic β-oxidation of $^{14}$C-labelled palmitic acid, a model fatty acid, was significantly enhanced by the addition of these three SCFAs into liver homogenates. As shown in Figure 6, γ-lactones are produced by the β-oxidation of middle- and/or long-chain fatty acids [9,11,26,27], a pathway of breaking down a long-chain acyl-CoA molecule to acetyl-CoA molecules to generate energy from lipids, followed by closing of the lactone ring. Thus, when β-oxidation of fatty acids in the liver is enhanced by SCFAs, the formation of γ-lactones can be promoted by an increase in blood γ-lactone levels, leading to elevated dermal emissions of sweet-smelling gases. This impression is reasonable because the blood route has been suggested as the most probable emission source for γ-lactones [28]. Therefore, the significant increase in the dermal emission flux of C10 and C11 lactones might be a result of the enhanced β-oxidation

of fatty acids mediated by SCFAs circulating in the blood after the ingestion of lactulose. The mechanism proposed here deserves to be verified in a more in-depth manner in further clinical studies.

**Figure 6.** A typical production pathway of γ-lactone from free fatty acid: γ-decalactone (C10) from palmitoleic acid with 3 steps of β-oxidation.

This study has a potential limitation in the study design; the new findings were based on open-label and before–after trials of lactulose. As an extended evaluation, a double-blind, placebo-controlled crossover trial must be conducted for investigating the prebiotic effect of lactulose on the dermal emission of sweet-smelling lactones.

## 5. Conclusions

The influence of ingestion of the food dosage level of lactulose was investigated on the cutaneous emissions of γ-lactones in healthy participants using a PFS-GC/MS methodology. For the first time, we discovered a significant increase in the fluxes of dermal emission of the sweet-smelling C10 and C11 lactones, as the number of bifidobacteria grew in the faeces, presumably mediated by SCFAs produced in the colon.

**Author Contributions:** Conceptualisation, Y.S. (Yoshika Sekine), Y.S. (Yohei Sakai), M.T. and H.O.; methodology, Y.S. (Yoshika Sekine), S.U., M.T., Y.S. (Yohei Sakai) and R.S.; investigation, Y.S. (Yoshika Sekine), S.U., M.T., Y.S. (Yohei Sakai) and R.S.; writing, Y.S. (Yoshika Sekine), M.T. and M.M., supervision, S.A., K.U. and H.O. All authors have read and agreed to the published version of the manuscript.

**Funding:** This research received no external funding. Y.S. (Yoshika Sekine), S.U., and M.T. have received a research grant from Morinaga Milk Industry Co., Ltd. Y.S. (Yohei Sakai), R.S., and H.O. were supported by Morinaga Milk Industry Co., Ltd. in the form of salaries.

**Institutional Review Board Statement:** This study was conducted in accordance with the Declaration of Helsinki. The study protocol was approved by the Institutional Review Board of the Shonan Campus, Tokai University (No. 18065).

**Informed Consent Statement:** Informed consent was obtained from all subjects involved in the study.

**Data Availability Statement:** All data generated or analysed during this study are included in this published article.

**Conflicts of Interest:** The authors declare no conflict of interest. The funder, Morinaga Milk Industry Co., Ltd., provided funding for authors, Y.S. (Yoshika Sekine), S.U., M.T., Y.S. (Yohei Sakai), R.S. and

H.O., but did not have any additional role in the study design, data collection and analysis, decision to publish, or preparation of the manuscript.

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
