# Peer review of "Influence of Ingestion of Lactulose on γ-Lactones Emanating from Human Skin Surface"

_applsci, doi:10.3390/app13063930_

Round 1
Reviewer 1 Report
In the manuscript “Prebiotic Effect of ingestion of lactulose on γ-lactones emanating from human skin surface” authors study the prebiotic effect of lactulose ingestion on the dermal emissions of γ-lactones in healthy participants after 2 weeks of 4 g/day lactulose consumption. They also studied the increment of bifidobacteria in faeces.
The work is very interesting and provides information regarding the consumption of lactulose with the release of volatile compounds through the skin and its correlation with the increase of bifidobacteria in faeces.
The paper is well written, concrete, concise and suitable for publication in the opinion of this referee. A few suggestions t authors:
Line 31 and 291: please replace “microflora” by “microbiota”
Line 44: please add a space before “[4,5]”
Lines 103-104: this referee do not understand the relevance of this affirmation: “Skin gas sampling was conducted for an additional 4 participants on Day 2 and 17 only.”
Lines 146-147: cell “numbers”? Or number?
Author Response
We greatly appreciate the constructive and useful recommendations. We have studied your comments carefully and made a serious effort to make appropriate changes as highlighted in the revised manuscript. Our answers to listed comments are described below. Please note after revising the manuscript following all reviewers’ comments, we rephrased some words to reduce an overlap with the previously published papers following the instruction by the editor.
Line 31 and 291: please replace “microflora” by “microbiota”.
(Answer) Respecting the comment, we have replaced “microflora” by “microbiota”.
Line 44: please add a space before “[4,5]”
(Answer) Thank you for your notice. We inserted the space in the sentence.
Lines 103-104: this referee do not understand the relevance of this affirmation: “Skin gas sampling was conducted for an additional 4 participants on Day 2 and 17 only.”
(Answer) Reviewer’s confusion may be due to our lack of explanation. The supplemental test was carried out to establish the relationship between dermal emissions of γ-lactones and the number of bifidobacteria. So, the skin gas sampling was conducted only on Day 2 and Day17, the days when faecal samplings were conducted. So, we have revised the sentence as below.
Line 104-105 “As for the additional 4 participants, skin gas sampling was conducted on Day 2 and 17 according to the date of faecal sampling.”
Lines 146-147: cell “numbers”? Or number?
(Answer) Since we applied the PCR methodology, we have deleted “cell” in the words and retained “numbers”.
We hope that the revised manuscript is now acceptable for publication.

Reviewer 2 Report
This manuscript with the title "Prebiotic Effect of ingestion of lactulose on g-lactones emanating from human skin surface" by Yoshika Sekine et al. describes the development of various human skin volatiles and the abundance of fecal Bifidobacterium during an intervention with lactulose.
Unfortunately, this work include some major flaws and I cannot recommend this manuscript for publication.
1) This works describes a human interventional study with 10 subjects (although the number is not fully clear to me, see comment 3 below) that have received daily doses of lactulose. The study design is very weak and does not allow reliable conclusions about the effects of lactulose (see e.g., Aggarwal and Ranganathan, doi:10.4103/picr.PICR_91_19). The major flaw of this design is the lack of a control group receiving either no lactulose or a suitable placebo substance. Any observed effect might as well be random or caused by uncontrolled factors depending directly on the intervention. Even the knowledge of being part of an intervention can change behaviour of participants, thereby causing all kinds of effects.
Thus, the conclusions drawn on the effects of lactulose treatment on Bifidobacterium and skin volatiles are not scientifically sound.
Other major points:
2) An ethics statement is missing. Human interventional studies should follow the Declaration of Helsinki on Ethical Principles for Medical Research Involving Human Subjects. Usually, the study needs to be approved by an ethics committee before any participant is enrolled.
3) The number of participants is not clear. Were there 6 participants (L. 75), 10 (L. 78) or 14 (L. 104: "additional 4 participants")
4) The authors declare no conflict of interest. However, Ryo Sakiyama is part of Morinaga Milk Industry Co., which provided lactulose and performed some of the measurements. Since the company presumably also sells lactulose, they have a financial interest in the outcome of the study. This must be declared.
5) Sampling and storage of stool samples prior to their arrival in the lab are not described (L. 147). However, this might influence the outcome of stool-related measurements.
6) The numbers of Bifidobacterium are given as "CFU per g feces", which is not appropriate. With qPCR methods, only the number of 16S rRNA gene copies can be determined. Bifidobacterium species in the samples might have different rRNA gene copy numbers in their genomes than the reference B. longum (see database rrnDB: https://rrndb.umms.med.umich.edu/). This could be amended by determining the actual species and their copy numbers. Or by measuring the abundance as "16S rRNA gene copies per g feces". In addition, CFU does not equal cell numbers, sinve cells might form aggregates, which can also differ between species, and even growth conditions.
7) In 2023, it seems somewhat outdated to only determine the abundance of a single fecal genus, since NGS methods (especially 16S rRNA amplicon sequencing) are quite affordable. Prebiotic treatment might substantially change many species' abundances in the microbiome and many bacteria contribute to the production of SCFA and other compounds.
8) The conclusion that the intervention "improved the gut microflora" (in the abstract and conclusions sections) seems a bit far fetched with only a single genus measured. Please do not use the term microFLORA. This is outdated and should be replaced by microbiota, microbiome or similar terms.
Author Response
We greatly appreciate the constructive and useful recommendations. We have studied your comments carefully and made a serious effort to make appropriate changes as highlighted in the revised manuscript. Our answers to listed comments are described below. Please note after revising the manuscript following all reviewers’ comments, we rephrased some words to reduce an overlap with the previously published papers following the instruction by the editor.
1) The study design is very weak and does not allow reliable conclusions about the effects of lactulose. The major flaw of this design is the lack of a control group receiving either no lactulose or a suitable placebo substance.
(Answer) Thank you very much for your valuable comment. We clearly understand the weakness in the study design, so we have added the following descriptions at the end of discussion.
Line 288-291: “This study has a potential limitation in the study design; the new findings were based on the open-label and before-after trials of lactulose. As an extended evaluation, a double-blinded and placebo-controlled study must be conducted for further strengthening the evidence obtained here.”
2)An ethics statement is missing. Human interventional studies should follow the Declaration of Helsinki on Ethical Principles for Medical Research Involving Human Subjects.
(Answer) I feel that this comment results from an oversight. We had described the Institutional Review Board Statement (Line 306-308) following the journal’s format.
3) The number of participants is not clear. Were there 6 participants (L. 75), 10 (L. 78) or 14 (L. 104: "additional 4 participants")
(Answer) As also pointed out by another reviewer, the description of L.104 in the original manuscript invited the confusion of reviewers. So, we have revised the sentence as below.
Line 104-105 “As for the additional 4 participants, skin gas sampling was conducted on Day 2 and 17 according to the date of faecal sampling.”
4) The authors declare no conflict of interest. However, Ryo Sakiyama is part of Morinaga Milk Industry Co., which provided lactulose and performed some of the measurements. Since the company presumably also sells lactulose, they have a financial interest in the outcome of the study. This must be declared.
(Answer) As we have clearly stated in the manuscript, this research received no external funding. Therefore, all authors do not have any conflict of interests.
5) Sampling and storage of stool samples prior to their arrival in the lab are not described (L. 147). However, this might influence the outcome of stool-related measurements.
(Answer) Thank you very much for this valuable comment. Respecting the comment, we have revised the descriptions.
Line 98-100 “The collected samples were immediately sent in cold conditions (-18℃) to the laboratory of Morinaga Milk Industry and stored at -80℃ until the use for real-time polymerase chain reaction (PCR).”
6) The numbers of Bifidobacterium are given as "CFU per g feces", which is not appropriate. the abundance should be expressed as "16S rRNA gene copies per g feces".
(Answer) Thank you again for this useful recommendation. Respecting this comment, we have replaced “CFU” by "16S rRNA gene copies per g faeces" or “GCNs per g faeces”.
7) In 2023, it seems somewhat outdated to only determine the abundance of a single fecal genus, since NGS methods (especially 16S rRNA amplicon sequencing) are quite affordable. Prebiotic treatment might substantially change many species' abundances in the microbiome and many bacteria contribute to the production of SCFA and other compounds.
(Answer) We clearly understand this valuable comment. Meanwhile, as shown in our previous report(*), no significant changes were observed in the abundance of high-occupancy intestinal bacteria other than Bifidobacterium on lactulose ingestion at the dose of 4 g/day. So, this study focused on the Bifidobacterium.
* Sakai, Y.; Ochi, H.; Tanaka, M. Lactulose Ingestion Induces a Rapid Increase in Genus Bifidobacterium in Healthy Japanese: a randomised, double-blind, placebo-controlled crossover trial. Microorganisms 2022, 10, 1719.
https://doi.org/10.3390/microorganisms10091719
8) Do not use the term microFLORA. This is outdated and should be replaced by microbiota, microbiome or similar terms.
(Answer) Respecting the comment, we have revised.
We hope that the revised manuscript is now acceptable for publication.

Reviewer 3 Report
The manuscript is given an overview of the effect of the consumption of lactulose prebiotic on the increase of bifidobacterial numbers and the synthesis of γ-lactones and their emission through the skin. This study is a part of systematic research dealing with the effect of lactulose consumption on the metabolism of colonic microbiota. The focus is on increasing the number of Bifidobacterium and the secretion of volatile metabolites through human skin.
The results presented in the manuscript contribute to the knowledge of the interaction between the colonic microbiota and the human body.
Author Response
We are grateful for the high evaluation on our paper that greatly encourages us.
Please note after revising the manuscript following all reviewers’ comments, we rephrased some words to reduce an overlap with the previously published papers following the instruction by the editor.

Round 2
Reviewer 2 Report
Please see the attachment

Author Response
We greatly appreciate the constructive and useful recommendations for the revised manuscript. We made a serious effort to make appropriate changes as highlighted in the revised manuscript. Our answers to listed comments are described below. Numbers corresponds to the first your review.
1)Main point of critique
(Answer) We understand the point of your criticism by carefully reading the additional comment. As you pointed out, we agree that the study design is too weak to establish the prebiotic effect of lactulose. So, we have made major revisions as follows.
First, we have deleted following statements you raised in the comment: “daily intake of 4 grams of lactulose improved the gut microbiota” (L31 in second version), “the number of bifidobacteria increased owing to the ingestion of lactulose” (L225 in second version), and “daily intake of lactulose improved the gut microbiota” (L295 in second version).
Second, we have revised the purpose of this study which investigated the “influence” of lactulose on the dermal emissions of γ-lactones, because it is difficult to prove the prebiotic effect of lactulose in the present study design. Accordingly, the title was changed to “Influence of ingestion of lactulose on γ-lactones emanating from human skin surface”.
Third, we have revised the limitation statement as follows: “As an extended evaluation, a double-blind, placebo-controlled crossover trial must be conducted for investigating the prebiotic effect of lactulose on the dermal emission of sweet-smelling lactones.” (L285-286)
4) Conflicts of interest
(Answer) We have asked some COI management committees in the field of medical science and engineering on the point. We are confused that the methods of disclosure seem to vary by an academic field at present. So, respecting you comment, we have added the declarations in Funding and Conflicts of Interest.
Funding: This research received no external funding. Y.SE, S.U., M.T. have received a research grant from Morinaga Milk Industry Co., Ltd.. Y. SA, R. S. and H.O. were supported by Morinaga Milk Industry Co., Ltd. in the form of salaries.
Conflicts of Interest: The authors declare no conflict of interest. The funder, Morinaga Milk Industry Co., Ltd. provided for authors, Y.SE., S.U., M.T., Y. SA, R.S. and H.O., but did not have any additional role in the study design, data collection and analysis, decision to publish or preparation of the manuscript.
Some minor revisions
We have revised all points following the comments.
We hope that the revised manuscript is now acceptable for publication.
